# A light-regulated host–guest-based nanochannel system inspired by channelrhodopsins protein

Yue Sun[1], Junkai Ma[2], Fan Zhang[1], Fei Zhu[1], Yuxiao Mei[1], Lu Liu[1], Demei Tian[1] & Haibing Li[1]

The light-controlled gating of ion transport across membranes is central to nature (e.g., in protein channels). Herein, inspired by channelrhodopsins, we introduce a facile non-covalent approach towards light-responsive biomimetic channelrhodopsin nanochannels using host–guest interactions between a negative pillararene host and a positive azobenzene guest. By switching between threading and dethreading states with alternating visible and UV light irradiation, the functional channels can be flexible to regulate the inner surface charge of the channels, which in turn was exploited to achieve different forms of ion transport, for instance, cation-selective transport and anion-selective transport. Additionally, the pillararene-azobenzene-based nanochannel system could be used to construct a light-activated valve for molecular transport. Given these promising results, we suggest that this system could not only provide a better understanding of some biological processes, but also be applied for drug delivery and various biotechnological applications.

[1] Key Laboratory of Pesticide and Chemical Biology (CCNU), Ministry of Education, College of Chemistry, Central China Normal University, Wuhan 430079, China. [2] Department of Chemistry, School of Pharmacy, Hubei University of Medicine, Hubei Key Laboratory of Wudang Local Chinese Medicine Research, Shiyan, Hubei 442000, China. Correspondence and requests for materials should be addressed to H.L. (email: lhbing@mail.ccnu.edu.cn)

Light-activated channels provide a precise and noninvasive optical means for controlling transport of ions across membranes, which is central to nature (e.g., in protein channels)[1-4]. For instance, channelrhodopsins (ChR), light-gated ion channels that serve as sensory photoreceptors, would function as a signal transmission medium to maintain normal functions of organisms[5-7]. Thus, the research on ChR is crucial for understanding their biological function in living cells. However, the fragility of the embedded lipid bilayers makes it hard to be applied in changing external environments, which limits their practical applications.

Fortunately, artificial nanochannels provide a good platform for tackling this challenge because of their greater flexibility in size, superior robustness, and surface properties[8-20]. To date, the scientific community has developed many smart nanochannels that can be responsive to molecules or ions owing to changing surface properties of the channels. The chief way of making nanochannels intelligent is through introduction of specific molecules. For biological ChR channels, photoinduced conformation transformation, natural ChR(originally cation-conducting) can be converted into chloride-conducting anion channels using the depolarization and hyperpolarization process[2, 5]. Chemically speaking, the design of functional molecules that can achieve this behavior and be used as building

**Fig. 1** Schematic of the design of a biomimetic light-activated nanochannel using host–guest systems inspired by ChR channel. The copyright belong to the Shenyang Zhiyan Technology Co., Ltd

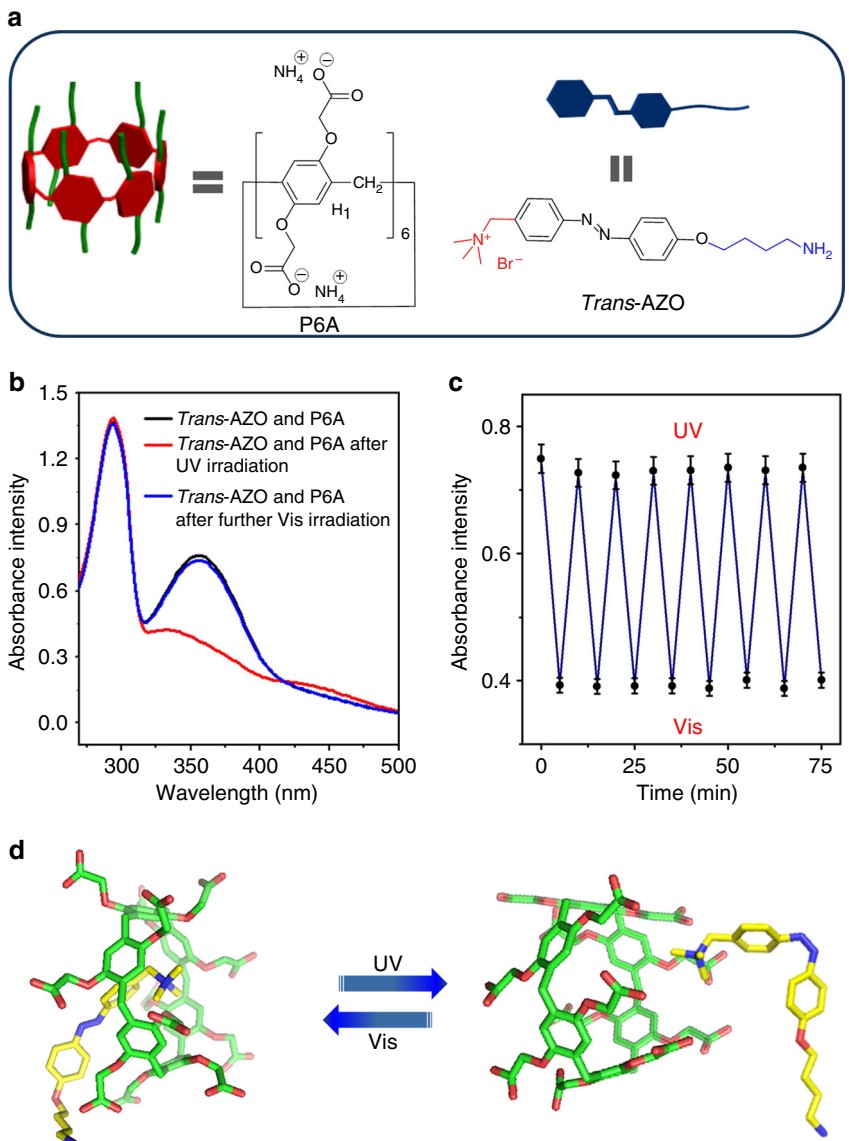

**Fig. 2** Host–guest interaction between P6A and AZO. **a** Schematic of host P6A and guest trans-AZO. **b** UV-Vis spectra of an equimolar solution of $1.00 \times 10^{-4}$ M AZO and P6A. **c** Changes in the absorbance at 345 nm of an equimolar solution of P6A and AZO upon alternating irradiation with UV and visible light. **d** Molecular stimulation of P6A and AZO by irradiation with UV and visible light

blocks for biomimetic light-activated channel systems is very important.

Host–guest systems, are good candidates for fabricating light-activated biomimetic channels because they encompass reversible non-covalent interactions between macrocyclic hosts and suitable guests[21, 22]. Host–guest systems consist of two components that can be separated into two individual components with charge separation in response to external stimuli[23]. Therefore, they could be used to achieve charge reversal in the internal surface of nanochannel, which is key to effective gating and selective transport of ion. As a new type of macrocyclic host, P6A has been intensively used as a molecular switch, similar to pseudorotaxanes, catenanes, and supramolecular dimers[24–31].

Herein, inspired by ChR channels, we describe a technique where host–guest systems were used to fabricate photosensitive artificial nanochannel. For this purpose, the inner wall of the nanochannel is primarily decorated with the positively charged azobenzene (AZO) molecules. The negatively charged pillararene (P6A) may form the inclusion compounds with AZO based on supramolecular self-assembly. The threading/dethreading transition between the AZO guest and P6A host could be controlled reversibly by using light irradiation as an external stimulus. This would lead to the inner surface charge change, which takes charge of the selective transport of ion species across the channels. Additionally, the P6A-AZO-based nanochannel system could be used to construct a light-activated valve for molecular transport. We anticipate that the integration of host–guest systems into nanostructures could not only provide a better understanding of some biological and pathological processes, but also have potential value for controlled drug release and various biotechnological applications (Fig. 1).

## Results

**Host–guest interaction betweenP6A and AZO.** Bearing this idea in mind, we designed and synthesized negative P6A and positive AZO (Supplementary Figs. 1 and 2). Ultraviolet and visible (UV-Vis) absorption spectroscopy was employed to confirm the photocontrollable threading–dethreading switch (Fig. 2a). As shown in Fig. 2b, the absorption band of the equimolar solution of P6A and *trans*-AZO at approximately 370 nm clearly decreased after irradiation under 365 nm UV light for 15 min. The decreased absorption peak was due to the isomerization of the *trans*−*cis*−AZO molecules. But under 435 nm irradiation for 15 min, the absorption peak intensity at approximately 370 nm could be basically returned to the original state. When an equimolar solution of P6A and *trans*-AZO were irradiated alternately with the light at 365 and 435 nm, this photoinduced *trans*–*cis* isomerization process could be cycled repeatedly (Fig. 2c).This demonstrated that the alternated UV and visible light irradiation could regulate reversibly the threading−dethreading transition because of the *trans*−*cis* photoisomerization of AZO. Subsequently, the nuclear magnetic resonance (NMR) titration experiments were used to determine the association constant ($K_a$) and stoichiometry of this host−guest complex. These data were identical with the reported literature[32, 33]. To further confirm photocontrollable threading–dethreading behavior, $^1H$ NMR characterization was conducted to provide evidence about the interaction of AZO with P6A. As shown in Supplementary Fig. 4, once an equimolar amount of P6A was added to free *trans*-AZO, the signal of protons on *trans*-AZO showed significant upfield shift. Moreover, the chemical shifts of protons Hb*, Hc*, and Hd* on *cis*-AZO benzene group changed slightly under UV light irradiation at 365 nm, suggesting that the protons Hc*, Hb*, and Hd* on *cis*-AZO benzene group was located outside the cavity of P6A (Supplementary Fig. 4D). The *cis*-AZO went back to *trans*-AZO under light

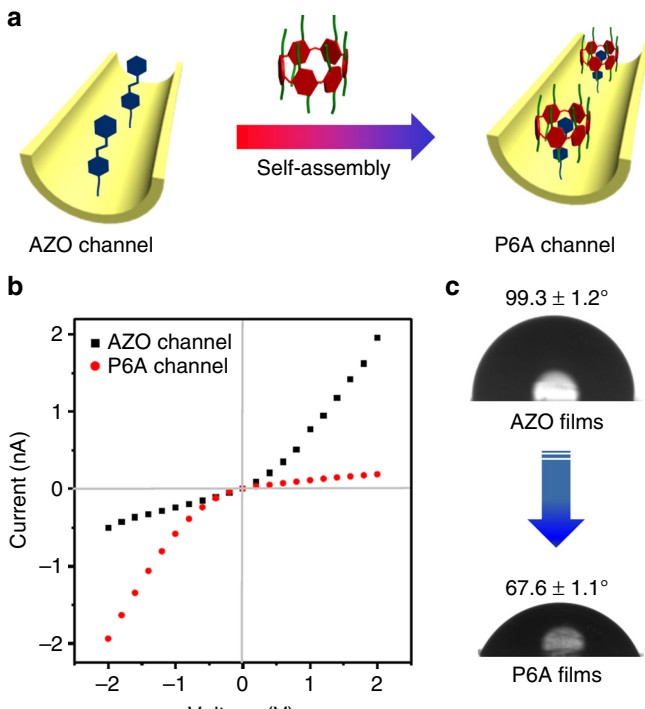

**Fig. 3** Fabrication of light-controlled nanochannels. **a** The construction process of light-controlled nanochannels using a host–guest system. **b** Current/voltage (*I/V*) curve change in the single nanochannel after each modification. **c** Change in contact angle on the PET films after each modification. These results indicated that the P6A-based light-sensitive system was coupled to the inner surface of the nanochannel

irradiation at 435 nm, and the proton signals in the solution of P6A and AZO restored to the initial state (Supplementary Fig. 4F), indicating that the optical control assembly–disassembly system between P6A and AZO could be realized. Subsequently, a molecular stimulation was conducted to further confirm the above results (Fig. 2d). These findings provided convincing evidence for the formation of the inclusion complex between P6A and AZO, mainly driven by electrostatic and hydrophobic interactions.

**Fabrication of light-controlled nanochannels using a host–guest system.** On the basis of these studies, a light-activated biomimetic nanochannel was readily fabricated by supramolecular self-assembly. Figure 3a shows the fabrication and operating process of the intelligent nanochannel. Firstly, the single conical nanochannel was prepared by asymmetric chemical etching of single-track polyethylene terephthalate membranes (PET) (Supplementary Fig. 6). The structure of the conical nanochannel was thoroughly studied with scanning electron microscope. The large opening diameter of the conical nanochannel was approximately 600 nm, and the small opening at the other side was approximately 20 nm (Supplementary Fig. 7). The carboxyl groups (–COO⁻) on the nanochannel surface were generated in the process of chemical etching condition, and the amino terminal in AZO was linked with the –COO⁻ groups using the classical 1-ethyl-3-(3-dimethyllaminopropyl) carbodiimide (EDC)/N-hydroxysulfosuccinimide (NHS) coupling reaction. Furthermore, the P6A molecules interact with AZO on the surface of the nanochannel by self-assembly (Fig. 3a, details in the experimental section). Hence, a light-controlled nanochannel was introduced onto the internal surface of the nanochannel by a simple two-step reaction. The *I*–*V* curve of the nanochannel changed after each modification (Fig. 3b).The original conical nanochannel has

negative charge due to the existence of ionized carboxyl groups and it strongly rectified the ionic current (Supplementary Fig. 9). After the modification of AZO on the nanochannel, the transmembrane ionic current at a positive voltage increased significantly due to the changes of the surface charge and surface wettability. Similarly, the current was increased at negative voltage after P6A was immobilized on the nanochannel, which could have been caused by the surface charge of P6A. To some extent, the changes in the $I-V$ curve demonstrated successful fabrication of the biomimetic channel.

To further underscore the successful construction of the biomimetic light-controlled channel, contact angle (CA) measurements, and X-ray photoelectron spectroscopy (XPS) analysis were carried out. As depicted in Fig. 3c, the wettability of the channel changed dramatically after the introduction of P6A. The CA of the etched membrane with unreacted $-COO^-$ was 71.6° ± 1.2° (Supplementary Fig. 10); after modification of the AZO compound, the CA of the functionalized surface increased to 99.3° ± 1.2°, resulting from the hydrophobicity of AZO, which was one of the reasons for the decrease in current for the AZO-modified nanochannel, as discussed above. After the immobilization of P6A, the CA decreased to 67.6° ± 1.1° owing to the hydrophilicity of P6A. In addition, the XPS analysis showed that the original PET films did not contain nitrogen; the peak for elemental nitrogen appeared after modification with P6A and AZO (Supplementary Fig. 11). All of the above data indicated that the P6A-modified nanochannels were successfully constructed by self-assembly.

**The ion selectivity of light-controlled nanochannels**. Subsequently, we investigated the photoswitching behavior of the system by recording transmembrane currents. The transmembrane currents were obtained in a 0.1 M KCl solution (pH = 7.03). Figures 4a, b reveal the current change of the identical host–guest-based nanochannel system before and after UV light irradiation.

Not unexpectedly, the smart nanochannel is cation selective because of the negative P6A host, and thus manifests rectification of the cationic channel. The prior direction of the potassium ions flux is from the small opening to the large opening. Upon UV irradiation of the P6A-based channel, the dethreading transition occurred owing to the *trans−cis* photoisomerization of AZO, resulting in the generation of positive AZO functionalities on the channel surface. The inner surface charge of the nanochannel was converted from negative to positive. It exhibited the anion selectivity along with inversion of the rectification features[34, 35]. Fortunately, the $I-V$ curves also exhibited the dramatic recovery during several cycles of threading/dethreading transitions (Fig. 4b).The electrical behavior of the nanochannel can be adjusted by a light-regulated host–guest-based nanochannel system via using the interactions between the charged internal surface and the mobile ionic species in solution.

**The switching between threading and dethreading states in the nanochannel**. To further confirm switching between threading and dethreading states in the nanochannel, the fluorescence of the nanochannel is observed in site by laser scanning confocal microscopy. As shown in Supplementary Fig. 12, we decorated the P6A with rhodamine B amine (RhB-NH₂) by condensation reactions in order to synthesize the fluorescent compound (P6A-RhB). The light-regulated host–guest systems were introduced into porous nanochannels by the noncovalent interactions between AZO and P6A-RhB. When the P6A-RhB assembled on the AZO-immobilized nanochannel, the nanochannel exhibited a strong fluorescence signal. The fluorescence thickness was ca. 13.0 ± 0.5 μm, which agreed with the actual thickness of the PET membrane. Subsequently, the functional nanochannel was further irradiated under the UV light. The fluorescence in the nanochannel weakened, which is likely to provide further evidence of the release of P6A.

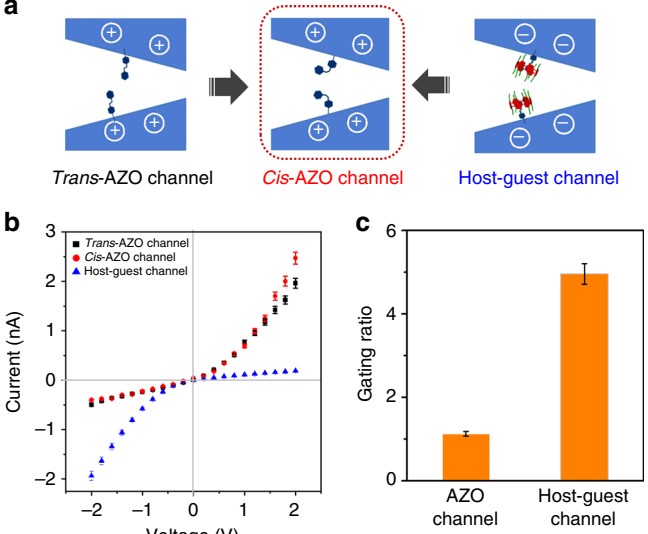

**Fig. 5** The advantage of light-regulated nanochannels on the basis of host-guest system. **a** Schematic illustration of the light-responsive channel obtained through AZO modification. **b** The *I–V* curve of the light-sensitive channel: light responsive AZO-modified channel and light responsive host–guest systems modified channel. **c** The light-driven ionic gating properties: gating ratios (*Rg*) of the nanochannel calculated by the ionic current measured before and after UV light irradiation. Standard deviation is ± 5% and are used for describing the *error bars*. Each data in two cases are tested five times, respectively

**Fig. 4** The ion selectivity of light-controlled mechanisms. **a** Analysis via current recording of the nanochannels before UV light irradiation. **b** Analysis *via* current recording of the nanochannels after UV light irradiation. **c** The reversibility of the different states of the P6A-based nanochannels by measuring the current after alternating irradiation with different light

Control experiments were carried out to show the advantage of the light-regulated nanochannel on the basis of host–guest system. The $I$–$V$ curves of the single-component AZO-modified nanochannel were examined under the same condition (Fig. 5). The Fig. 5c shows the light response ratio (before UV irradiation/after UV irradiation ratio) histogram, which was calculated using the ratio between current before and after UV light irradiation at −2 V. The ratio of the AZO-modified nanochannel is approximately 1, whereas the ratio of host–guest modified systems increased sharply to 5. These findings indicate that light can be used to efficiently regulate the P6A-AZO-modified nanochannel from cation-selective to anion-selective. We further investigate the use of this photoresponsive nanodevice for regulating the transport of molecule cargo. Adenosine triphosphate (ATP) was chosen for the cargo because it was a negative molecule and played a key role in several major metabolic processes[36]. As shown in Supplementary Fig. 13, before UV light irradiation, the inner walls of the multichannel membrane were negative owing to the presence of the negative P6A moieties. The speed of ATP transport is very slow. However, the P6A was released after UV irradiation, leading to AZO groups being exposed. These moieties transformed the negative inner channel walls into positively charged channel wall. Thus, the permeation of ATP was significantly increased.

## Discussion

In conclusion, learning from the ChR channel in live cell, we developed functional nanochannels with capabilities of photo-controlled transport by integrating a photoswitchable host–guest system with biomimetic membranes. The AZO ⊂ P6A-modified channel system was reversibly and cyclically actuated by switching between threading and dethreading states using alternated visible and UV light. In particular, we characterized the optical gating of cargo transport using a model molecule (ATP). Given these promising results, we expect that the marriage of nanostructures and versatile devices could not only provide a better understanding of some biological processes, but could also be applied for drug delivery and various biotechnological applications.

## Methods

**The construction of conical nanochannel**. The conical nanochannel based on PET material was prepared through the ion track etching method. Before etching process, the PET films was irradiated with the UV light (365 nm) for 1 h on both sides. To obtain the conical nanochannel, alkali solution was added to one side, the other side of the device was provided with an acid solution that can neutralize the etching solution as soon as the channel opens. Specifically, the PET membrane was placed in the middle of the conductivity cell at constant 30 °C, which is composed of two chambers. The etching alkali solution (9 M NaOH) was added on one side, and the blocking solution (1 M KCl + 1 M HCOOH) on the other side. To observe the current signal, a constant 1 V was performed across the PET membrane. According to the requirement of the channel size, the etching process can be stopped at the suitable current. In order to remove the residual salts remaining within the channel, the membrane was washed by the pure water five times.

**The functionalization of nanochannel**. The inner surface of the nanochannel contains a large number of carboxyl group (–COOH) in the process of the etching PET membrane. In order to activate carboxyl groups, the PET membrane was immerged in the water solution containing EDC (15 mg)/NHS (3 mg). This state was maintained for 1 h at room temperature and then the membrane was washed with pure water three times. The pre-functionalized channel further interacted with 5 mM AZO for an overnight time period. Then, the AZO-modified channel was placed in P6A solution ($10^{-3}$ M) for 5 h. The functionalized channels can be fabricated successfully after washing with distilled water.

**Ion currents measurement**. Keithley 6487 picoammeter (Keithley Instruments, Cleveland, OH) could be used to measure the ion currents. Ag/AgCl electrodes were used to conduct a transmembrane potential across the membrane. The film was placed in the middle of the conductance cell. Both halves of the cell were filled with a 0.1 M KCl solution prepared. In order to record the $I$–$V$ curves, a scanning triangle voltage signal from −2 to +2 V was selected. Each test was repeated five

times to obtain the average current value at different voltages. Specifically, before exposure to UV radiation, the transmembrane currents were obtained in a 0.1 M KCl solution under a scanning triangle voltage signal from −2 to +2 V. Upon irradiation with UV light, the functional nanochannel was fixed in the halves of the cell. This process was supported further by applying a potential of +5 V on the side containing a 0.1 M KCl solution for 1 h. Then the PET film was immersed in methanol for 5 h. After that, the functionalized channels were further washed several times with distilled water. To measure the resulting ion current flowing through the nanochannel, a scanning voltage between −2 to +2 V on the two sides was applied.

**CA measurement**. Data of CA were taken through CA apparatus containing an OCA 20 system (Dataphysics, Germany) at room temperature. The original PET film was primarily etched with alkali solution (9 M NaOH) for 1 h. The PET film was taken out from the alkali solution and washed with a blocking solution (1 M KCl + 1 M HCOOH) for 30 min. Then PET film was further immerged in deionized water for 12 h. The PET film was then dried with $N_2$ before the CA experiment. The value of contact angel was picked up five times at the membrane surface by adding an approximately 2 μL droplet of water.

**X-ray photoelectron spectra experiment**. X-ray photoelectron spectra (XPS) data were obtained with an ESCALab220i–XL electron spectrometer from VG Scientific using 300 W Al $K_\alpha$ radiation. In this work, to further prove that the P6A and AZO modified successfully by measuring the nitrogen element, all peaks were referenced to C1s (CHx) at 284.8 eV in the deconvoluted high-resolution C1s spectra.

**Gaussian calculation**. Computational calculations were carried out at the density functional theory b3lyp/6-31 G (d) levels using Gaussian 03.

**Data availability**. The authors declare that the data supporting the findings of this study are available within the article and its Supplementary Information files and all relevant data are available from the authors.

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

## Acknowledgements

This work was financially supported by the National Natural Science Foundation of China (21572076, 21372092), Natural Science Foundation of Hubei Province (2013CFA112, 2014CFB246), Wuhan scientific and technological projects (2015020101010079).

## Author contributions

Y.S. and H.L. conceived and designed the experiments. Y.S. made a major contribution in all the experiments. Y.S., J.M., F.Z., F.Z, Y.M., L.L., D.T., and H.L. wrote the manuscript. Y.S. and H.L. supervised all the experiments and analyses.

## Additional information

**Competing interests:** The authors declare no competing financial interests.

