## [Peer Review File · Nature Communications]

Reviewers' comments:

Reviewer #1 (Remarks to the Author):

In this manuscript, the authors demonstrated an artificial channel system that used a facile non-covalent approach towards light-responsive biomimetic ChR nanochannels. The positive results indicate a quite successful preliminary attempt which might not only help promote the understanding of some biological processes, but also be applied for drug delivery and various biotechnological applications. Publication of this article is recommended after the authors have addressed the following comments:

(1) What is the source for the PET foil?

(2) Before the etching process, each side of the PET membrane was exposed to UV light (365 nm) for 1 hour." Please tell the significance of this process.

(3) "During the chemical etching process, carboxyl groups (-COO-) were exposed on the nanochannel surface." According to the structural formula of PET, please tell how could the carboxyl groups (-COO-) expose on the nanochannel surface.

(4) Details of the computational model (Fig. 1C) are lacking. The authors should give details about the type of model (e.g. QM, MD etc.), the corresponding parameters (e.g. basis sets) and the coordinates of the final molecular structures.

(5) Introduction: "This would lead to a change in the surface charge". Which surface? the inner surface or the external one? You should point it out.

(6) It would be nicer if this rationale can be elaborated to clearly demonstrate the underlining reason why pillar[6]arene is better than other receptors, such as pillar[5]arene.

(7) "Additionally, ¹H NMR titration experiment were used to determine the association constant (K_a)". The author should illustrate the meaning of association constant in NMR measurement.

(8) In page 2, Paragraph 1: " the fragility of the embed lipid bilayers makes it hard for them to be applied in changing external environments, which limits their practical applications" . Please delete "for them".

(9) In page 2, Paragraph 2, "the scientific community has developed many smart nanochannels that can be responsive to molecules or ions owing to changing surface properties of the attached on the channels". Please delete "the attached on".

(10) In page 2, Paragraph 2: "photoinduced conformation transformation, natural ChR (originally cation-conducting) may convert into chloride-conducting anion channels using a depolarization and hyperpolarization process". You may express it as "photoinduced conformation transformation, natural ChR (originally cation-conducting) can be converted into chloride-conducting anion channels using the depolarization and hyperpolarization process"

(11) In page 2, Paragraph 3: Host-guest systems consist of two components that can be separated into their two individual components with charge separation in response to external stimuli accompanied with charge separation. You may express it as "Host-guest systems consist of two components that can be separated into two individual components with charge separation in response to external stimuli"

(12) In page 2, Paragraph 3: "Herein" instead of "Here"

(13) In page 2, Paragraph 3: "inspired by ChR channels, we describe a novel technique in which that host-guest systems were used to..." "that" instead of "in which"

(14) In page 2, Paragraph 4: "approximately 370 nm clearly decreased after irradiation with 365 nm UV light at 365 nm for 5 min." You may express it as "approximately 370 nm clearly decreased after irradiation under 365 nm UV light for 5 min"

(15) In page 3, Paragraph 1: "using a 0.1 m KCl solution pre-pared in a 0.1 M KCl solution (pH = 7.03) on both sides of the membrane" Change "pre-pared" to "prepared".

(16) In page 4, Paragraph 3: "...learning from the ChR channel in a live cell, were developed by integrating a photoswitchable P6A-based host-guest system with biomimetic membranes" "in live cell" instead of "in a live cell". And delete "P6A-based"

Reviewer #2 (Remarks to the Author):

This manuscript reported a novel light-regulated host-guest-based nanochannel system by integrating a photoswitchable P6A-based host-guest system into biomimetic membranes. By using this photocontrolled threading-dethreading switch, the nanochannel system has the ability to control surface charge and further photoswitchable cation /anion selective transport can be constructed. UV-Vis absorption spectroscopy, ¹H NMR titration, current-voltage curves measurement and scanning electron microscopy were conducted to support the conclusions drawn by the authors. The manuscript is interesting. However, I think a minor revision is needed for several reasons.

1. The authors should provide more evidence to confirm the photocontrollable threading-dethreading behavior based on host-guest system of anionic pillar[6]arenes (P6A) and a positive azobenzene guest (AZO). In my opinion, only molecular stimulation (Figure 1C) confirmed photocontrollable threading-dethreading process in this article.
2. Upon irradiation with UV light, the trans-cis isomerization of the trans-AZO molecules occurred and cis-AZO molecule dethreaded of the cavity of P6A. However strong electrostatic interactions also existed between cis-AZO and P6A. P6A still adhered to cationic part of AZO (shown in Figure 1C), which indicated that the surface charge may have no changes. In Figure 3, I-V curves showed a different result. The authors should explain it.
3. In Figure 2C, the reviewer wanted to know the contact angle of P6A film after irradiation with UV light.
4. The title of reference 14a was wrong.
5. The following important paper related to pillararenes/azobenzene-based photocontrollable system should be cited in reference 14: J. Am. Chem. Soc., 2015, 137, 1440-1443.

Reviewers' comments:

Reviewer #1 (Remarks to the Author):

In this manuscript, the authors demonstrated an artificial channel system that used a facile non-covalent approach towards light-responsive biomimetic ChR nanochannels. The positive results indicate a quite successful preliminary attempt which might not only help promote the understanding of some biological processes, but also be applied for drug delivery and various biotechnological applications. Publication of this article is recommended after the authors have addressed the following comments:

(1) What is the source for the PET foil?

Answer: The PET foil is a kind of polymer film, and the ingredient of PET is poly(ethylene terephthalate) (as showing in the following structure).

PET

Figure 1. The ingredient of PET films

(2) Before the etching process, each side of the PET membrane was exposed to UV light (365 nm) for 1 hour." Please tell the significance of this process.

Answer: The treatment with the UV light leads to a saturation of the damage in the tracks, so that the further storage of the samples in air or illumination with visible light does not change the etching behavior.

(Apel, P. Y.; Korchev, Y. E.; Siwy, Z.; Spohr, R. & Yoshida, M. *Nucl. Instrum. Meth. B*, (184), 337-346, 2001)

(3) "During the chemical etching process, carboxyl groups (-COO⁻) were exposed on the nanochannel surface." According to the structural formula of PET, please tell how could the carboxyl groups (-COO⁻) expose on the nanochannel surface.

Answer: Then ingredient of PET films is polyethyleneterephthalate. And polyethyleneterephthalate was hydrolyzed in alkali condition. It may generate -COO⁻ (detailed in following picture)

PET

Figure 2. The carboxyl groups (-COO⁻) expose on the nanochannel surface during preparation.

(4) Details of the computational model (Fig. 1C) are lacking. The authors should give details about the type of model (e.g. QM, MD etc.), the corresponding parameters (e.g. basis sets) and the coordinates of the final molecular structures.

Answer: The binding of P6A and AZO were examined by computational calculations at b3Lyp/6-31G(d) levels by using Gaussian 03. And cartesian coordinates of P6A binding AZO can be found in supporting information.

(5) Introduction: "This would lead to a change in the surface charge". Which surface? the inner surface or the external one? You should point it out.

Answer: According to your suggestion, the surface charge means the inner surface. We corrected it.

(6) It would be nicer if this rationale can be elaborated to clearly demonstrate the underlining reason why pillar[6]arene is better than other receptors, such as pillar[5]arene.

Answer: Inspired by the ChR nanochannels, from a chemical point of view, the critical problem is that designing a negative host and a positive guest. Pillar[6]arene is easily synthesized and modified. For pillar[5]arene host, it can not complex with *trans*-AZO guest due to the cavity sizes of pillar[5]arene is smaller (it is demonstrated by the literature (*J. Am. Chem. Soc.* **2012**, **134**, 8711–8717)).

(7) "Additionally, ¹H NMR titration experiment were used to determine the association constant (K_a)". The author should illustrate the meaning of association constant in NMR measurement.

Answer: The association constant (K_a), it is associated with the binding and unbinding reaction of host (P6A) and guest (AZO) molecules. And it can show the binding affinity between P6A and AZO.

(8-16) The reviewer points out that many sentences are not grammatically correct.

Answer: We corrected our grammatical mistakes with your advice. And then we passed the article to the native speaker to further confirm the language correctness and language usage (blue part in the paper). And two examples are given out.

(8) In page 2, Paragraph 1: " the fragility of the embed lipid bilayers makes it hard for them to be applied in changing external environments, which limits their practical applications" . Please delete "for them".

Answer: As your kind suggest, we delete the "for them" (Blue part in page 2, paragraph 1)

(9) In page 2, Paragraph 2, "the scientific community has developed many smart nanochannels that can be responsive to molecules or ions owing to changing surface properties of the attached on the channels". Please delete "the attached on".

Answer: In page 2, Paragraph 2, we have deleted " the attached on".

Reviewers' comments:

Reviewer #2 (Remarks to the Author):

This manuscript reported a novel light-regulated host–guest-based nanochannel system by integrating a photoswitchable P6A-based host–guest system into biomimetic membranes. By using this photocontrolled threading–dethreading switch, the nanochannel system has the ability to control surface charge and further photoswitchable cation /anion selective transport can be constructed. UV-Vis absorption spectroscopy, ¹H NMR titration, current-voltage curves measurement and scanning electron microscopy were conducted to support the conclusions drawn by the authors. The manuscript is interesting. However, I think a minor revision is needed for several reasons.

1. The authors should provide more evidence to confirm the photocontrollable threading–dethreading behavior based on host–guest system of anionic pillar[6]arenes (P6A) and a positive azobenzene guest (AZO). In my opinion, only molecular stimulation (Figure 1C) confirmed photocontrollable threading–dethreading process in this article.

Answer: To confirm photocontrollable threading–dethreading behavior, ¹H NMR characterization was conducted to provide evidence about the interaction of *trans*-AZO and *cis*-AZO with P6A. Compared with free *trans*-AZO (Figure. S1A), significant chemical shift changes of the signals for the protons on *trans*-AZO occurred in the presence of an equimolar amount of P6A (Figure. S1B). The peaks related to Ha, Hb, Hc, Hd shifted upfield remarkably (-0.96, -0.48, 0.56, -0.68 ppm, respectively). Moreover, these peaks became broad owing to complexation dynamics. The reason for the extensive changes of the chemical shifts is that these protons are located within the cavity of P6A and are shielded by the electron-rich cyclic structure upon forming a threaded structure between P6A and *trans*-AZO. Additionally, the protons on P6A also exhibited chemical shift changes. The peak related to H₁ shifted downfield from 6.54 to 7.12 ppm. These evidences show the formation of an inclusion complex between P6A and *trans*-AZO (The following picture).

As shown in Figure S1E, the molar ratio of the *trans* to *cis* form of AZO changed to 50 : 50 after irradiation with UV light at 365 nm for 15 min. And the chemical shift of proton Ha* of *cis*-AZO shifted upfield from 7.01 to 5.41 ppm in the presence of equimolar P6A (Figure 1D). Respectively, and exhibited a broadening effect, suggesting the complexation between P6A and *cis*-AZO. Moreover, the chemical shifts of protons Hb*, Hc*, and Hd* on the benzene rings of *cis*-AZO changed slightly,

indicating that the benzene ring containing protons Hc*, Hb* and Hd* of guest *cis*-AZO was outside the cavity of P6A. However, upon irradiation with light at 435 nm for 15 min, *cis*-AZO went back to *trans*-AZO, and the proton signals related to the solution of P6A and AZO went back to the original state (Figure S1F), suggesting that the photo-controllable *threading–dethreading* switch between P6A and AZO was achieved.

Figure S1 Partial ¹H NMR spectra (400 MHz, D₂O, room temperature): (A) *trans*-AZO (3.0 mM); (B) *trans*-AZO (3.0 mM) and P6A (3.0 mM); (C) P6A (3.0 mM); (D) *trans*-AZO (3.0 mM) and P6A (3.0 mM) after irradiation at 365 nm for 15 min; (E) *trans*-AZO (3.0 mM) after irradiation at 365 nm for 15 min; (F) *trans*-AZO (3.0 mM) and P6A (3.0 mM) after further irradiation at 435 nm for 15 min.

2. Upon irradiation with UV light, the *trans–cis* isomerization of the *trans*-AZO

molecules occurred and *cis*-AZO molecule dethreaded of the cavity of P6A. However strong electrostatic interactions also existed between *cis*-AZO and P6A. P6A still adhered to cationic part of AZO (shown in Figure 1C), which indicated that the surface charge may have no changes. In Figure 3, *I*-*V* curves showed a different result. The authors should explain it.

Answer: Upon the irradiation with UV light, we speculated that the P6A would be released in the nanochannel. And the external voltage may further take the P6A to the electrolyte solution. Thus, the change of inner surface charge appears under a scanning voltage (as shown in Figure 3C).

To support our hypothesis, the fluorescence of the nanochannel is observed in site by laser scanning confocal microscopy (Figure S2). We used the P6A fluorescent derivative (P6A-RhB), which was synthesized by linking the amino group to the rhodamine B amine (RhB-NH₂). A host-guest complex was then formed on the AZO-modified porous PET membrane by the interaction between AZO and P6A-RhB. As shown in the following picture, when the P6A-RhB successfully assembled on the AZO-immobilized nanochannel, the nanochannel exhibited a strong fluorescence signal. The fluorescence thickness was ca. 13.0±0.5 μm, which agreed with the actual thickness of the PET membrane. Subsequently, the functional nanochannel was further irradiated under the UV light. The fluorescent in the nanochannel weakened, which is likely to provide further evidence of the release of P6A.

Figure S2. Laser scanning confocal microscopy (LSCM) images observed the fluorescence change of the nanochannel towards UV light irradiation

3. In Figure 2C, the reviewer wanted to know the contact angle of P6A film after irradiation with UV light.

Answer: According to your kind suggestions, the contact angle of P6A film under irradiation with UV light is supplemented with pertinent experimental data (Both in supporting information and the following picture). We hypothesize that the change of contact angle is due to the release of the P6A towards irradiation with UV light.

Figure S3. The contact angle of P6A-based host-guest PET films systems towards UV/Vis irradiation.

4. The title of reference 14a was wrong.

Answer: We corrected the title of reference 14a.

5. The following important paper related to pillararenes/azobenzene-based photocontrollable system should be cited in reference 14: *J. Am. Chem. Soc.*, 2015, 137, 1440–1443.

Answer: We benefit a lot from the reference (*J. Am. Chem. Soc.*, (137), 1440-1443, 2015). And we have cited the article.

REVIEWERS' COMMENTS:

Reviewer #1 (Remarks to the Author):

For the queries and review comments on this manuscript, corresponding authors have fully addressed with their clear responses and comments and even with a number of additional experiments in the response letter. Accordingly, I found the changes made in the revised manuscript is fully satisfactory. Therefore I would like to recommend this revised manuscript to be accepted for publication in Nature Communication as it is.